# Metabolic Profiling and Antioxidant Analysis for the Juvenile Red Fading Leaves of Sweetpotato

**DOI:** 10.3390/plants11223014

**Published:** 2022-11-08

**Authors:** Jie Shi, Qiang Wu, Jiliang Deng, Kelly Balfour, Zhuo Chen, Yonghua Liu, Sunjeet Kumar, Yanli Chen, Zhixin Zhu, Guopeng Zhu

**Affiliations:** 1Key Laboratory for Quality Regulation of Tropical Horticultural Crops of Hainan Province, School of Horticulture, Hainan University, Haikou 570228, China; 2Yazhou Nanfan Service Center of Agricultural and Rural Bureau, Sanya 572025, China; 3Department of Biology, Algoma University, Sault Sainte Marie, ON P6A 2G4, Canada

**Keywords:** sweetpotato leaves, anthocyanins, juvenile red fading, metabolite profiling, antioxidants

## Abstract

Leaves of sweetpotato (*Ipomoea batatas* L.) are promising healthy leafy vegetable. Juvenile red fading (JRF) leaves of sweetpotato, with anthocyanins in young leaves, are good candidates for developing functional vegetables. Here, metabolic profiling and possible antioxidants were analyzed for five leaf stages of the sweetpotato cultivar “Chuanshan Zi”. The contents of anthocyanins, total phenolics, and flavonoids all declined during leaf maturation, corresponding to declining antioxidant activities. By widely targeted metabolomics, we characterized 449 metabolites belonging to 23 classes. A total of 193 secondary metabolites were identified, including 82 simple phenols, 85 flavonoids, 18 alkaloids, and eight terpenes. Analysis of the metabolic data indicates that the antioxidant capacity of sweetpotato leaves is the combined result of anthocyanins and many other colorless compounds. Increased levels of “chlorogenic acid methyl ester”, a compromised form of chlorogenic acid, significantly correlated with the declined antioxidant abilities. Besides anthocyanins, some significant metabolites contributing to the high antioxidant property of the sweetpotato leaves were highlighted, including chlorogenic acids, isorhamnetin glycosides, *trans*-4-hydroxycinnamic acid methyl ester, 4-methoxycinnamic acid, esculetin, caffeate, and trigonelline. This study provides metabolic data for the utilization of sweetpotato leaves as food sources, and sheds light on the metabolomic change for JRF leaves of other plants.

## 1. Introduction

Juvenile red fading (JRF) is the phenomenon in plants whereby red young leaves gradually turn green as they mature, due to alternate accumulation of anthocyanins and chlorophylls [1,2]. Anthocyanins are strong antioxidants, and red young leaves are often consumed as nutritional food with antioxidant properties, such as the young leaves of Chinese toon (*Toona sinensis*) [3]. JRF occurs frequently in woody plant species, but scarcely in herbaceous plants [4,5]. The consumption of young leaves on trees is highly seasonal and causes damage to trees.

Sweetpotato (*Ipomoea batatas* L.) is an important herbaceous food crop in Asia, with the tuberous roots the major consumable tissue. Tender leaves and stems of sweetpotato are also highly nutritious [3,6], and are increasingly accepted as health-promoting vegetable by consumers, possibly due to chlorogenic acid (5-*O*-caffeoylquinic acid, 5-CQA) and its derivatives [6,7,8,9,10,11]. The extract from sweetpotato leaves was reported to have higher antioxidant activity than the majority of other common vegetables [3,12]. Powders processed from sweetpotato leaves were recommended as functional ingredients in food products [7,8]. Fresh sweetpotato leaves are indicated to be potent dietary antioxidants in the prevention of inflammations, tumors, diabetes, and many other cardiovascular diseases [6,9,10].

Among the rich color patterns displayed by sweetpotato varieties, purple leaves with anthocyanins are reported to have stronger antioxidant abilities than green leaves, possibly due to their higher anthocyanin, quercetin, and polyphenol levels [3,6,13]. However, purple mature leaves are often associated with taste of astringency and are much less welcomed by consumers. While red young leaves are adorable and attractive, additional anthocyanin accumulation in young leaves but not in mature leaves could further increase their health value and avoid problems concerning the taste of astringency. Sweetpotato varieties with JRF phenomenon would be ideal candidates for developing functional leafy vegetable.

“Chuanshan Zi” is a widely distributed purple-fleshed sweetpotato cultivar having tender leaves at tips of the stems which display obvious JRF phenomenon, and it has great value for developing health vegetable. We explored its pigment change and underlying molecular mechanism for anthocyanin accumulation in our former research [2]. Here, to further explore the change of antioxidant ability and underlying metabolites occurring in sweetpotato JRF leaves, the dynamic changes in metabolite accumulation were explored with the JRF leaves across five developmental stages of “Chuanshan Zi” (C1, C2, C3, C4, and C5). We first analyzed the changes of anthocyanins, total phenolic content (TPC), total flavonoid content (TFC), and the antioxidant abilities. Then, by LC-ESI-MS/MS (liquid chromatography-electrospray ionization- tandem mass spectroscopy)-based widely targeted metabolomic analysis [14], we reported the large-scale identification and quantification of metabolites for the five leaf stages. The correlation coefficients were calculated between the antioxidant activities and the identified 23 metabolite classes and the 193 secondary metabolites. Some possible significant metabolites contributing to the high antioxidant property of sweetpotato leaves were highlighted. This study sheds light on the metabolic changes occurring in plant JRF leaves, and provides metabolic data for the utilization of sweetpotato leaves as food sources.

## 2. Results

### 2.1. The Antioxidant Capacities for Juvenile Red Fading Leaves of “Chuanshan Zi”

Leaves of “Chuanshan Zi” turned from red to green across five developmental stages: C1, C2, C3, C4, and C5 (Figure 1A), corresponding to the alternate accumulation of anthocyanins and chlorophylls during JRF [2]. Total phenolics and flavonoids, important indicators of the antioxidant capacities of vegetables, decreased from C1 to C5 (Figure 1B–D). The decrease of TPC across the five stages was mild, from 5.90 to 3.88 mg GAE/gFW. The decrease in TFC was more drastic, dropping from 54.06 to 8.75 mg rutin/gFW.

Three antioxidant indices were assessed, including the scavenging of 2,2′-diphenyl-1-picrylhydrazyl (DPPH), the scavenging of 2,2′-azino-bis (3-ethylben-zothiazoline-6-sulphonicacid) (ABTS·+), and the ferric reducing power (Figure 1E). As collectively indicated by the three indices, the antioxidant capacities also showed a descending order of C1 > C2 > C3 > C4 > C5. However, it should be noted that the antioxidant capacities of C4 and C5 were very close.

### 2.2. Metabolic Profiling for Juvenile Red Fading Leaves of “Chuanshan Zi”

Metabolomic data mainly refers to endogenous small molecules with a relative molecular weight less than 1000 [14]. Through LC-ESI-MS/MS based widely targeted metabolomics, a total of 449 metabolites belonging to 23 classes were identified from the five leaf stages (Table 1 and Appendix A). Among them, 233 compounds (Class 1–8) represent primary metabolites, which were mainly the intermediate products of primary metabolism. Class 1–4 (i.e., Amino acids and their derivatives, Nucleotides and their derivatives, Lipids, and Carbohydrates) represent components of the four major categories of biomolecules (i.e., proteins, nucleic acids, lipids, and sugars). Other small primary metabolites included Class 5 (Organic acids and their derivatives), Class 6 (Vitamins and their derivatives), Class 7 (Indole derivatives), and Class 8 (Alcohols).

Secondary metabolites are mainly grouped into three major categories: phenolics, alkaloids, and terpenoids. Here, a total of 18 alkaloids (Class 9), eight terpenoids (Class 10), and 167 phenolics (Class 11–22) were detected (Table 1). Phenolics are generally divided into two smaller categories: simple phenols and flavonoids. Simple phenols contain phenylpropanoids and their modified compounds, and here we identified 82 simple phenols (Class 11–14). Chlorogenic acid (5-CQA) and its derivatives were the main reported phenolics of sweetpotato leaves [7,13]. Here, 5-CQA, 4-CQA, 3,4-diCQA, 1,3-diCQA, and chlorogenic acid methyl ester were identified (Appendix A; Class 12, Phenolic acids). Classification of flavonoids is more diverse, including flavanones, flavones, isoflavones, flavonols, anthocyanins, proanthocyanidins, and polyphenols; and here we identified 85 flavonoids (Class 15–22). Among them, five anthocyanin compounds were identified (Class 20, Anthocyanins), all belonging to peonidin/cyanidin-derived glucosides, consistent with earlier reports [2,6].

### 2.3. Accumulation Trends for the 23 Metabolite Classes

Based on the multiple reaction monitoring (MRM) mode during MS/MS analysis (Appendix A), relative quantities of metabolites were calculated as the chromatographic peak area integrals (Appendix A). Principal component analysis (PCA) of the 449 compounds clearly separated the leaf samples of five stages (Figure 2A), where PC1 and PC2 accounted for 38.75% and 16.1% of the variation, respectively. The quality control samples (mix 01–03) were closely clustered in the center of the diagram, indicating the reliability of the experiment.

By adding up the peak areas, the relative compound abundance of the 23 classes was determined (Figure 2B). The classes not belonging to phenolics (Class 1–10, and Class 23) are shown in the upper panel of Figure 2B. Class 1–5 had the highest summed peak areas, representing the abundance of the four basic primary metabolites (Class 1–4), and organic acids and their derivatives (Class 5). Phenolic compounds (Class 11–22) are shown in lower panel of Figure 2B. The peak areas of Class 14 (Quinones) and Class 18 (Isoflavones) were very low, indicating trace quantities of the two classes. Notably, although there were only five anthocyanin compounds, their summed peak area of Class 20 was significant, reflective of the high anthocyanin contents in the red young leaves of “Chuanshan Zi”.

Relative accumulation trends for metabolites of the 23 classes were shown by standardized peak areas (mean peak area of each stage/mean peak area of the five stages) (Figure 2C). Within each class, the general trend (shown by the red line) was the same as that in Figure 2B, while the trends of individual compounds (shown by the gray lines) varied greatly. The Class 12 (Phenolic acids, highlighted by yellow shade), containing the main reported functional CQA phenolics, showed a combined upward trend, contrary to the gradual decline in antioxidant activities along leaf development (Figure 1E), possibly due to effects of some individual compounds. For Class 20 (Anthocyanins, highlighted by pink shade), the five individual anthocyanins have similar trends: C2 > C1 > C3 > C4 > C5, with the highest content in C2. Then, anthocyanin content dropped significantly from C2 to C3, consistent with fading of the red color in mature leaves.

### 2.4. Differential Metabolites between Adjacent Leaf Stages

With the criteria of |log_2_FC| ≥ 1 and VIP ≥ 1, the differential metabolites (DMs) between adjacent leaf stages were screened. A total of 170 substances accumulated differentially in at least one of the four comparison groups (C1-vs-C2, C2-vs-C3, C3-vs-C4, and C4-vs-C5). The 170 DMs belong to all classes, except Class 14 (Quinones) and Class 18 (Isoflavones), which had only trace quantities indicated by their low peak areas. The status of DMs is shown in volcano plots (Figure 3A). There were 60 DMs in C1-vs-C2 (30 up and 30 down), 84 DMs in C2-vs-C3 (31 up and 53 down), 41 DMs in C3-vs-C4 (15 up and 26 down), and 74 DMs in C4-vs-C5 (13 up and 61 down). C4-vs-C5 group contained the highest proportion of down-DMs, indicating that many metabolites have higher contents in young leaves than in mature leaves.

The top 10 significant up- and down-DMs in each comparison group are shown (Appendix A), corresponding to the most discrete red and green dots in the volcano plots (Figure 3A). The compounds marked with * had |log_2_FC| > 8, mainly due to the fact that they were absent in one of the leaf stages. The other DMs had values of |log_2_FC| < 5, including lipids, flavonoids, phenylpropanoids, phenolics, etc. (Appendix A).

The top 20 significantly enriched KEGG pathways for DMs were analyzed for the four comparison groups (Figure 3B). Significant differences were shown for pathways associated with phenolics: Pathways of phenylpropanoid biosynthesis (ko01061 or ko00940) appeared in all four comparison groups; pathway of flavonoid biosynthesis (ko00941) appeared in the first three groups; and pathway of anthocyanin biosynthesis (ko00942) appeared in the KEGG analysis of C2-vs-C3 and C4-vs-C5, reflective of the drastic decline of red pigment during these stages.

### 2.5. Analysis of the Top 50 Abundantly Accumulated Compounds

By ranking the mean peak areas of the five stages for each compound, the 449 compounds were given their rank orders of abundance (Appendix A), and the top 50 abundant compounds were shown (Table 2). 

For the top 50 abundant compounds, their average peak areas were from 8.47 × 10^7^ to 1.36 × 10^7^. The peak area values over 10^7^ continued until the 73th compound, and the 100th compound had a mean peak area of 7.10 × 10^6^. Calycosin (Class 18, Isoflavones) had a peak area of 2.70 × 10^3^ and ranked the last.

Among the 20 most abundant compounds, there were 17 primary metabolites (mainly from Class 1, 3, and 5) and three secondary metabolites. “Cynaroside (luteolin 7-*O*-glucoside)” is the only flavone in Table 2, but it ranked No.1 among all compounds and accounted for 59.2% of the total peak area of Class 17 (Flavones). “Chlorogenic acid methyl ester” (Class 12, Phenolic acids) ranked No.17 and is a derivative of chlorogenic acid (5-CQA). “*Trans*-4-hydroxycinnamic acid methyl ester” (Class 11, Phenylpropanoids) ranked No.18 and is an important precursor in phenolic metabolism.

Among the metabolites from No.21 to No.50, there were 17 secondary metabolites, including 16 phenolics and one alkaloid (trigonelline, No.32). Moreover, four of the five identified anthocyanins were included (No.23, 33, 36, and 40), indicative of the high anthocyanin contents in the red young leaves of “Chuanshan Zi”.

Among the top 50 compounds, 16 were DMs for adjacent leaf stages. Three of them, i.e., No.18 (*trans*-4-hydroxycinnamic acid methyl ester), No.41 (4-methoxycinnamic acid), and No.35 (caffeate), also appeared in Appendix A as among the top 10 significant DMs (marked with triangles). No significant changes were found for the other 34 compounds during leaf development, indicating that they are relatively stable substances within sweetpotato leaves. These data could serve as an important reference for the nutritional composition of sweetpotato leaves.

### 2.6. Correlation between Antioxidant Capacities and Metabolites

Correlation coefficients (R) between the antioxidant capacities and the 23 metabolite classes were calculated (Table 1, Appendix A). Among the primary metabolites, Class 2 (Nucleotide and derivatives) and Class 8 (Alcohols) had significant positive correlations with antioxidant capacities (R > 0.9), while Class 5 (Organic acids and derivatives) showed a significant negative correlation (R < −0.9). Among the secondary metabolites, Class 11 (Phenylpropanoids), Class 13 (Phenolamides), and Class 21 (Proanthocyanidins) were significantly positively correlated with antioxidant capacities (R > 0.9), while Class 12 (Phenolic acids) and Class 14 (Quinones) were negatively correlated (R < −0.9). The R values of the 193 individual secondary metabolites (including 167 phenolics, 18 alkaloids, and eight terpenes) were further calculated to show contributions of individual metabolites (Appendix A).

Class 20 (Anthocyanins, R = 0.779) is key to leaf color change during JFR. The accumulation of the five anthocyanins were shown with their rank and R values (Figure 4A). They had similar accumulation trends that declined from C2 to C5, and all had positive R values (from 0.461 to 0.877), with “Peonidin 3-*O*-sophoroside-5-*O*-glucoside” the most prominent.

Class 12 (Phenolic acids, R = −0.966) contains the main reported antioxidant-related compounds for sweetpotato leaves—chlorogenic acid (5-CQA) and its derivatives. The significant negative R value of Class 12 was unexpected, but became reasonable after we scrutinized its members. Among the 25 members, five CQA derivatives were included (Figure 4B). The negative R value of Class 12 was largely due to the high proportion of “chlorogenic acid methyl ester” (No.17, R = −0.9413), which is the second most abundant secondary metabolite (Table 2).

With |R| > 0.9 as the threshold, 46 secondary metabolites significantly correlated with antioxidant capacities (33 with R > 0.9, and 13 with R < −0.9) (Appendix A, in red). Nine of them were also among the top 50 abundant substances (Table 2), including one compound with R< −0.9 (Figure 4B, chlorogenic acid methyl ester) and eight compounds with R > 0.9 (Figure 4C). The eight significantly positive compounds include three isorhamnetin glycosides (isorhamnetin 3-*O*-glucoside, isorhamnetin *O*-hexoside, and isorhamnetin 5-*O*-hexoside), four simple phenols (*trans*-4-hydroxycinnamic acid methyl ester, caffeate, 4-methoxycinnamic acid, and esculetin), and one alkaloid (trigonelline), and their contents declined in the process of leaf development. Peak areas of the three isorhamnetin glycosides exceeded 10^7^ in the stage of C5, although they were much lower than in C1. For the four simple phenols and one alkaloid, their peak areas were high in early stages but dropped below 10^7^ at C3 or later stages, indicating that they are more specific to the youngest leaves. Moreover, notably, Cynaroside (Class 17, Flavone; No.1; R = −0.7517), the compound with the highest mean peak area, increased from C2 to C5, and showed very high levels in mature leaves. It is the glycoside of Luteolin (Class 17, No.271; R = 0.9383), which had high positive correlation with antioxidant capacities (Figure 4D).

## 3. Discussion

### 3.1. The Metabolic Profile for Leaves of Sweetpotato “Chuanshan Zi”

Metabolic profiling for leaves of “Chuanshan Zi” revealed 449 metabolites, and 233 (accounting for 51.9%) were primary metabolites representing components of four major categories of biomolecules (e.g., proteins, nucleic acids, lipids, and sugars). Moreover, 170 compounds accumulated differentially in at least one of the adjacent leaf stage comparison groups. The remaining 279 compounds showed no significant change during leaf development. These compounds, especially those also among the top 50 abundant metabolites, can be used as reference for the nutritional composition of sweetpotato leaves.

Of the 193 secondary metabolites identified, 86.5% can be classified as phenolics, one of the three major categories of secondary metabolites. Phenolics are widely distributed in plants and are reported to have various physiological functions. Among them, chlorogenic acid and its derivatives were the main reported functional components with strong antioxidant property in sweetpotato leaves [6,7,8,9,10,11]. Chlorogenic acid is often reported as 3-CQA [7,11], but is actually 5-CQA according to the IUPAC nomenclature system [13,15].

In addition, special attention should be paid to another member of the phenolics: Cynaroside (luteolin 7-*O*-glucoside; No.1; R = −0.7517), the compound with the highest mean peak area. It can be hydrolyzed to luteolin by *β*-glucosidase and is a more stable storage form of luteolin that possesses lower antioxidant capacity [16,17]. In our data, the instability and efficiency of luteolin (No.271, R = 0.9383) can be shown by its low accumulation levels, high SE variance, and high correlation with antioxidant capacities. Cynaroside showed high accumulation levels in all five stages, especially in older leaves at stages C4 and C5. This accumulation mode of cynaroside in mature leaves may lead to the rapid production of luteolin under adverse conditions.

### 3.2. Key Compounds Responsible for Declining Antioxidant Capacities in Leaves of “Chuanshan Zi”

Decreased antioxidant capacities were detected for JRF leaves of “Chuanshan Zi”. In our screening of key responsible metabolites, the abundance of compounds was considered, and nine significantly correlated prominent compounds were found.

Among these nine compounds, the only negatively correlated compound was “chlorogenic acid methyl ester” (No.17; R = −0.9413), the second most abundant secondary metabolite and the most abundant CQA derivative in our metabolic data. Mono-, di-, and tri-*O*-CQAs were confirmed to have strong antioxidant capacities [6,11]. Here, five forms of CQAs were identified in the leaf blade, with 3,4-diCQA (No.57; R = 0.8903) the second most abundant. Chlorogenic acid methyl ester is the methylated derivative of chlorogenic acid (5-CQA) and may also function as the more stable storage form of CQAs, that can promptly degrade to CQAs in mature leaves.

The other eight screened compounds have R > 0.9 and were highly expressed in young leaves and then declined during leaf development. These eight compounds include three isorhamnetin glycosides, four simple phenols, and one alkaloid, and were all reported to be functional phytochemicals. “Isorhamnetin glycosides” were demonstrated to had higher DPPH radical scavenging ability than ascorbic acid [18]. “Caffeate”, “*trans*-4-hydroxycinnamic acid methyl ester”, and “4-methoxycinnamic acid” are all derivatives of cinnamic acid and have been attributed to similar pharmacological effects as CQAs, such as antioxidants, antimicrobials, and protection of the liver and heart [19,20,21,22]. By replacing the hydroxyl group of *trans*-4-hydroxycinnamic acid with a methoxyl group, “4-methoxycinnamic acid” was derived, although with weakened antioxidant capacities [20]. “Esculetin (6,7-dihydroxycoumarin)”, a bioactive coumarin derivative, also possesses various pharmacological activities against obesity, diabetes, renal failure, thrombosis, and cardiovascular disorders [23]. Esculetin was determined to be an effective radical scavenging substance under UV radiation and was recommended to be used in cosmetics to prevent photo-aging [24]. For the alkaloid “trigonelline”, studies showed it results from the detoxification of excessive nicotinic acid, and it can play a key role in subcellular energy metabolism as an NAD reserve [25,26]. The above studies also agree that trigonelline mainly accumulated in juvenile tissues, such as embryos and young leaves, which is in accordance with our data.

Although many other substances were also identified as significant in the DM analysis or correlation analysis, we did not choose them as key substances due to their lower peak areas. However, their possible special functions cannot be ruled out. Schemes for the decreased antioxidant abilities in sweetpotato JRF leaves were summarized (Figure 5).

### 3.3. Anthocyanins in Juvenile Leaves May Have Multiple Functions Relating to Both the Red Color and Their Antioxidant Capacities

Juvenile red leaves are aesthetically appealing and are also potential functional foods due to their prominent anthocyanin contents. In this study, the five identified anthocyanins had similar accumulation trends, which declined from C2 to C5 during the leaf development of “Chuanshan Zi”. Their contents were relatively high in the red leaves. Three of these anthocyanins were among the 170 DMs in the comparison groups of C2-vs-C3 and C4-vs-C5, consistent with the results of our KEGG enrichment analysis.

Anthocyanins are important antioxidants. The five identified anthocyanins all had positive R values (from 0.461 to 0.877), rendering a positive R value (R = 0.779) for Class 20. However, these values were not the most significant among the 193 secondary metabolites. Our data showed that there are abundant colorless antioxidants within sweetpotato leaves, such as simple phenols and other flavonoids as discussed above. It has been reported that kaempferol (a flavonol) and chlorogenic acid accumulated in immature leaves and decreased in mature leaves, thereby protecting young tissues from natural UV radiation [27,28]. Although the exact function of anthocyanins in juvenile leaves is still under debate [5], two compelling explanations are both related to the red color of anthocyanins. One explanation is that the pigments act as light attenuators through the absorption of high-energy blue-green light [4], a wavelength spectrum that cannot be effectively absorbed by colorless antioxidants (although they can absorb UV light). The other explanation concerns the vision of insects: that the eyes of most insects lack red photoreceptors and have week red color perception, which can alleviate insect herbivory of juvenile red leaves [29,30]. Combined with our data regarding sweetpotato leaves, it is likely that anthocyanins in leaves can provide diverse protective roles, and anthocyanins in juvenile leaves may have multiple functions relating to both the red color and their antioxidant capacities.

## 4. Materials and Methods

### 4.1. Plant Materials

Sweetpotato were grown in fields located in the agricultural base on campus of Hainan University, which is in the tropical coastal city of Haikou. To avoid variation caused by external environmental factors, standard growth conditions were adopted: After applying organic fertilizer to the soil, healthy vines were replanted in new field plots every three months from February to October, from the year 2017 to 2022. Plants were subjected to natural light conditions and temperatures from 15℃ to 35℃. After approximately 1.5–2 months of growth under such adequate nutrition and light, the leaves at tips of the stems would be fresh and tender, with the color change from red to green clearly seen on each vine. The five leaf developmental stages were the same as those described by Deng et al. [2], with the apical tip of the vine defined as C1, and the leaves further along the vine defined as stages C2 to C5 (Figure 1A).

Leaf samples for this study were collected between May and July 2020. Typical vines were collected in the field between 10 a.m. and 11 a.m. to avoid the influence of circadian rhythm. For each stage, leaves from five vines were combined as one sample, quickly frozen in liquid nitrogen, and then stored at –80 ℃ for later usage. Each sample was ground into a fine powder with liquid nitrogen before usage. Three samples were tested as biological repeats for each of the subsequent assays.

### 4.2. Measurement of Anthocyanins, TPC, TFC, and Antioxidant Activities

Anthocyanins were extracted with 1% HCl-methanol and measured according to Luo et al. [1] and Sun et al. [31]. Total anthocyanin content (mg·g^−1^ FW) = (A_530_ − 0.25 × A_657_) × V × DR × MW/(ε × L × m). In the above formulas, V is the extraction volume (mL); DR is the dilution rate; MW is the molecular weight (449.2 g mol^−1^ for cyanidin-3-glucoside); ε is the molar extinction coefficient (29,600 L mol^−1^·cm^−1^ for cyanidin-3-glucoside); L is the path length (1 cm) of the cuvette; and “m” is the fresh weight (FW).

For measurements of the total phenolic contents (TPC) and total flavonoid contents (TFC), about 1 g of ground leaf sample powder was homogenized in 10 mL of 75% ethanol. Supernatants were collected after centrifugation (4 °C, 10,000× *g* for 1 min). The TPC was measured using the Folin-Ciocalteu reagent (Guangzhou chemical reagent factory, Guangzhou, China), and the results were evaluated at 760 nm and expressed as milligrams of catechin equivalents per gram of leaf powder (mg GAE/gFW). The total flavonoids (including flavones, isoflavones, flavonols, and dihydroflavonols) were measured using sodium nitrite-aluminum nitrate as in the work of Li et al. [32]. The TFC was evaluated at 508 nm, and the results were expressed as milligrams of rutin equivalents per gram of leaf powder (mg rutin/gFW).

Three antioxidant indexes were assessed according to the methods of Liu et al. [33], including the scavenging of 2,2′-diphenyl-1-picrylhydrazyl (DPPH·), the scavenging of 2,2′-azino-bis (3-ethylben-zothiazoline-6-sulphonicacid) (ABTS·+), and the ferric reducing power. Leaves were extracted with 75% ethanol using the same process as measurements for TPC and TFC. Through linear fitting of the scavenging rate curve, values of IC50 (concentration corresponding to 50% scavenging of radicals) were obtained. Lower IC50 values correspond to higher antioxidant activities. The reciprocal values of IC50 were used for subsequent calculation of correlation coefficients. 

Data from three biological replicates were collected for each of the above measurements. The means and standard error (SE) were presented.

### 4.3. Sample Preparation for Metabolomic Analysis

A total of 15 leaf samples for the five stages of “Chuanshan Zi” were independently freeze-dried under vacuum, with each stage containing three biological replicates. Then, 100 mg of the dried powder was extracted overnight at 4 °C with 1.0 mL 70% aqueous methanol. Following centrifugation at 10,000× *g* for 10 min, the extracts were absorbed (CNWBOND Carbon-GCB SPE Cartridge, 250 mg, 3 mL; ANPEL, Shanghai, China, www.anpel.com.cn/cnw) and filtered (SCAA-104, 0.22 μm pore size; ANPEL, Shanghai, China, http://www.anpel.com.cn/) before LC-MS analysis.

Three quality control (QC) samples were also prepared to assess the repeatability of the results. Equal volumes of extracts from each of the 15 samples were mixed and then divided into three equal aliquots to obtain the QC samples (mix 01–03). The QC samples were injected after every five experimental samples throughout the analytical run to provide a set of data from which repeatability could be assessed.

### 4.4. LC-ESI-MS/MS Analysis

The leaf sample extracts were analyzed by LC-ESI-MS/MS system (HPLC, Shim-pack UFLC SHIMADZU CBM30A system, www.shimadzu.com.cn/; MS, Applied Biosystems 6500 Q TRAP, www.appliedbiosystems.com.cn/). 

The LC and MS/MS conditions were performed and determined as described by Chen et al. [14]. Briefly, Waters ACQUITY UPLC HSS T3 C18 (1.8 µm, 2.1 mm*100 mm) were used as the HPLC column. The effluent was alternatively connected to an ESI-triple quadrupole-linear ion trap (Q TRAP)-MS. Linear ion trap (LIT) and triple quadrupole (QQQ) scans were acquired on the Q TRAP-MS. QQQ scans were acquired as multiple reaction monitoring (MRM) mode experiments with collision gas (nitrogen) set to 5 psi. Declustering potential (DP) and collision energy (CE) for individual MRM transitions were completed with further DP and CE optimization. A specific set of MRM transitions was monitored for each period according to the metabolites eluted within this period.

### 4.5. Qualitative and Quantitative Analysis of Metabolites

Qualitative analysis of primary and secondary MS data was carried out by comparison of the accurate precursor ions (Q1), product ions (Q3), the retention time (Rt), and fragmentation patterns to those obtained by injecting standards using the same conditions if the standards were available (Sigma-Aldrich, Waltham, MA, USA, http://www.sigmaaldrich.com/united-states.html) or conducted using the self-compiled database MWDB (MetWare biological science and Technology Co., Ltd., Wuhan, China). Repeated signals of K^+^, Na^+^, NH_4_^+^, and other large molecular weight substances were eliminated during identification.

Quantitative analysis of metabolites was based on the MRM mode (Appendix A). The characteristic ions of each metabolite were screened using the QQQ mass spectrometer to obtain the signal strengths. Integration and correction of chromatographic peaks was performed using MultiQuant version 3.0.2 (AB SCIEX, Concord, Ontario, Canada). The corresponding relative metabolite contents were represented as chromatographic peak area integrals. For the substances not expressed in certain stages (e.g., 6-gingerol and *N-p*-coumaroyl agmatine were not found in C1), the peak areas were recorded as 9 to allow for statistical analysis (Appendix A).

Unsupervised principal component analysis (PCA) was carried out using the statistics function prcomp within R v3.5.0 (www.r-project.org). Supervised multiple regression orthogonal partial least squares-discriminant analysis (OPLS-DA) was used to obtain values of the variable importance in project (VIP). Compounds with VIP ≥ 1 and |Log_2_FC| ≥ 1 (fold change; FC ≥ 2 or FC ≤ 0.5) were screened as differential metabolites (DMs) in each comparison group.

### 4.6. Statistical Analysis

All data are presented as the means and standard error (SE) of three biological replicates. Analysis of variance was performed using the SPSS software. Differences were tested by one-way ANOVA and Duncan’s test, and differences of *p* < 0.05 were taken to be statistically significant. Pearson’s bi-directional analysis was used to obtain the correlation coefficients (R) between metabolites and the antioxidant capacities along the five leaf stages. 

## 5. Conclusions

This study provides large-scale metabolic data for JRF leaves of sweetpotato “Chuanshan Zi”. Decreased antioxidant capacities were detected for JRF leaves during development, consistent with decreased contents of anthocyanins, total phenolics, and total flavonoids. Metabolic data indicate that the antioxidant capacities of sweetpotato leaves are the combined result of anthocyanins and many other colorless compounds. Decreased antioxidant abilities for JRF leaves are mainly attributed to: (1) Decreased antioxidant metabolites, including anthocyanins, CQAs, isorhamnetin glycosides, *trans*-4-hydroxycinnamic acid methyl ester, 4-methoxycinnamic acid, esculetin, caffeate, trigonelline, and so on; and (2) increased metabolites with compromised antioxidant abilities, such as cynaroside (compromised form of luteolin) and chlorogenic acid methyl ester (compromised form of 5-CQA). Anthocyanins in juvenile leaves may have multiple functions relating to both the red color and the antioxidant capacities. This study furthered our understanding of the metabolite changes of JRF leaves of sweetpotato. Future studies on the functional verification of these compounds, or the study of other plants would help us to reach more decisive conclusions.

## Figures and Tables

**Figure 1 plants-11-03014-f001:**
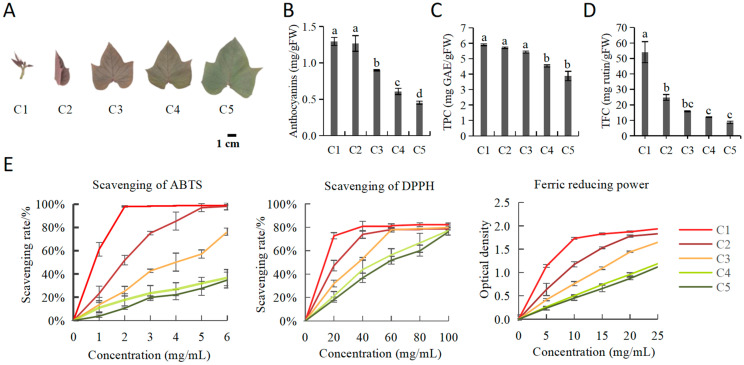
The contents of anthocyanins, total phenolics, total flavonoids, and antioxidant abilities for the five leaf stages of “Chuanshan Zi”. (**A**) The five leaf developmental stages from C1 to C5. The contents of (**B**) anthocyanins, (**C**) total phenolics (TPC), (**D**) total flavonoids (TFC) across the five leaf stages. (**E**) Antioxidant activities indicated by ABTS/DPPH radical-scavenging activity and ferric reducing power. Different letters mean significant differences (*p* < 0.05). All values are expressed as means ± SE, n = 3.

**Figure 2 plants-11-03014-f002:**
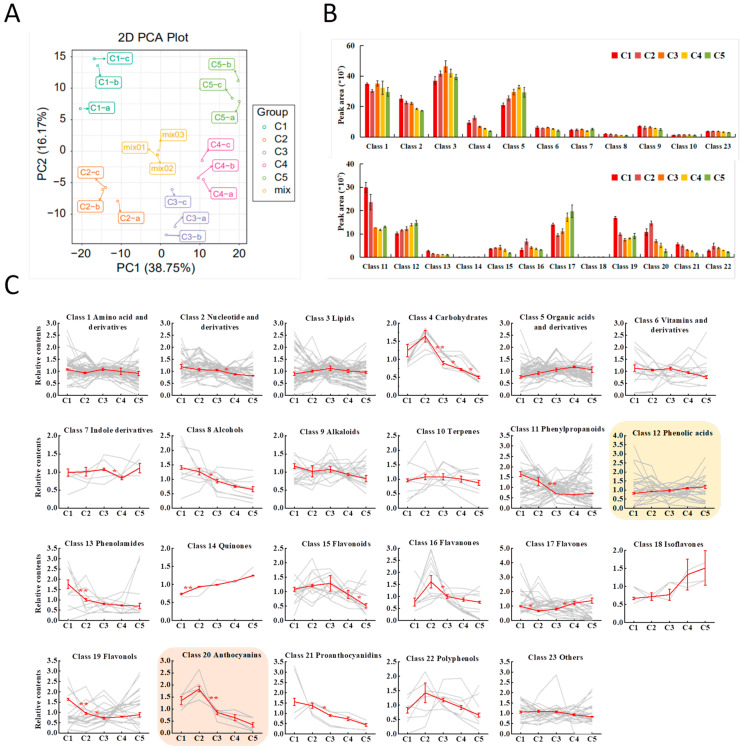
Metabolite accumulation for the 23 classes in leaf stages from C1 to C5. (**A**) PCA analysis of the 449 identified metabolites. Mix 01–03 were the three quality control samples. (**B**) Metabolite abundance of the 23 classes indicated by the added-up peak areas of compounds within each class. (**C**) Relative accumulation trends for compounds of the 23 classes. The trends were reflected by standardized peak areas (mean peak area of each stage/mean peak area of the five stages). Each gray line represents the trend for one individual compound. The red lines represent the combined trends of the compounds from each class. Class 12 (Phenolic acids) and Class 20 (Anthocyanins) were highlighted by yellow and pink shades, respectively. All values are expressed as means ± SE, n = 3. The significant differences for adjacent stages are indicated by asterisks at *p* < 0.05 (*) and *p* < 0.01 (**).

**Figure 3 plants-11-03014-f003:**
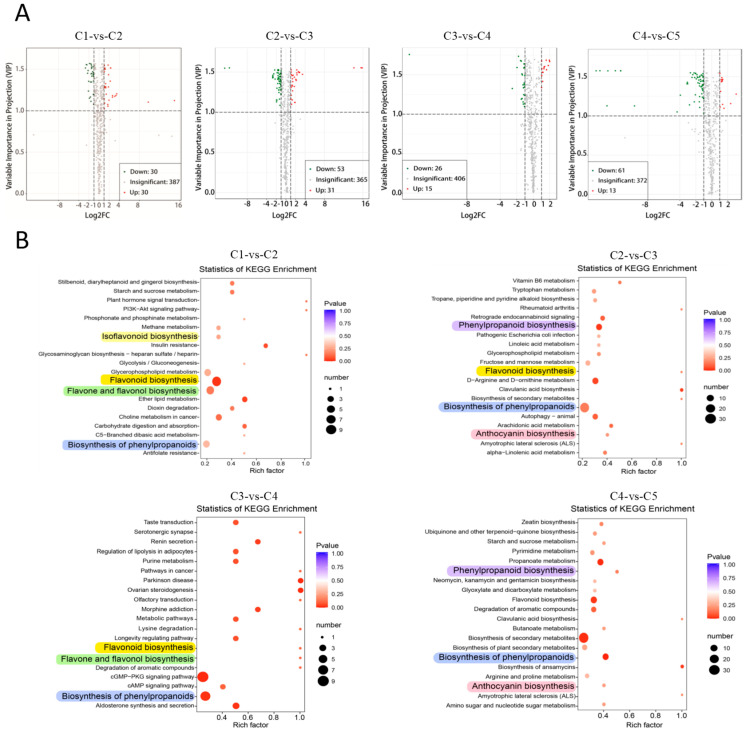
Differential accumulating metabolites for adjacent leaf stages. (**A**) Volcano plots of the identified DMs for adjacent leaf stages, with the red, green, and gray dots showing up-DMs, down-DMs, and insignificant DMs, respectively. (**B**) Top 20 significantly enriched KEGG pathways for adjacent leaf stages. The *X*-axes indicate the “enrich factor” represented by the ratio of DMs numbers to total annotated compound numbers of each pathway. *Y*-axes on the left represents KEGG pathways. The area of a circle represents the DM number. The KEGG pathways associated with biosynthesis of phenolics were highlighted.

**Figure 4 plants-11-03014-f004:**
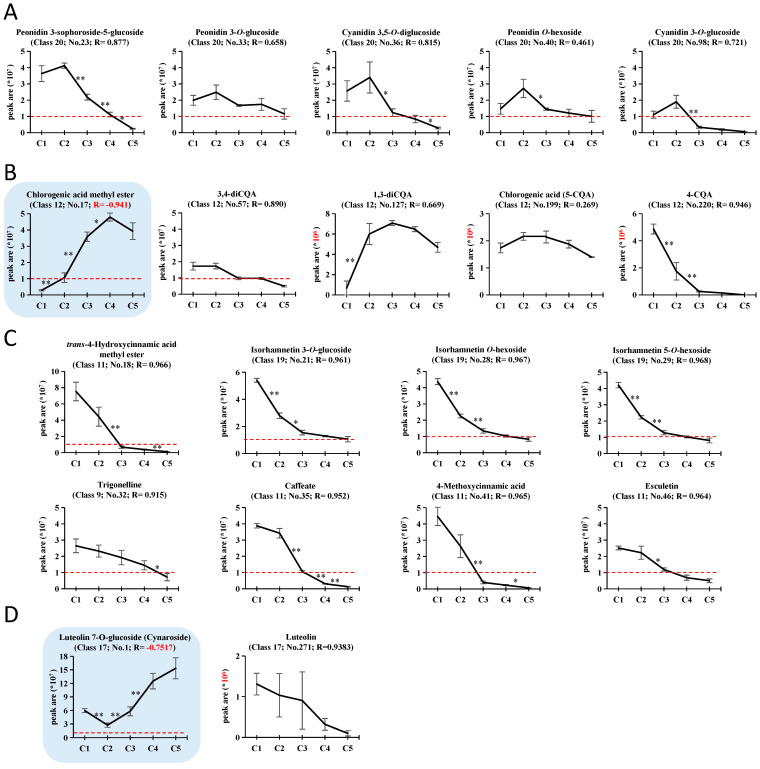
The accumulation of some possible functional compounds relating to the antioxidant capacities. (**A**) The five anthocyanins from Class 20, which are key to leaf color change during JFR. (**B**) Chlorogenic acid (5-CQA) and its derivatives from Class 12. Among the five metabolites, “chlorogenic acid methyl ester” was the only one within the top 50 abundant compounds with R < −0.9, and was highlighted by blue shade. (**C**) The eight compounds within the top 50 abundant compounds with R > 0.9. (**D**) Cynaroside and luteolin. Cynaroside had negative R, and was highlighted by blue shade. The class, ranking number, and R value are shown for each compound. Red dashed lines represent the abundance peak area of 1 × 10^7^. The *Y*-axes for luteolin and the three CQAs not exceeding 1 × 10^7^ were indicated in red. Values are expressed as means ± SE, n = 3. The difference significance for adjacent stages are indicated by * (*p* < 0.05) and ** (*p* < 0.01).

**Figure 5 plants-11-03014-f005:**
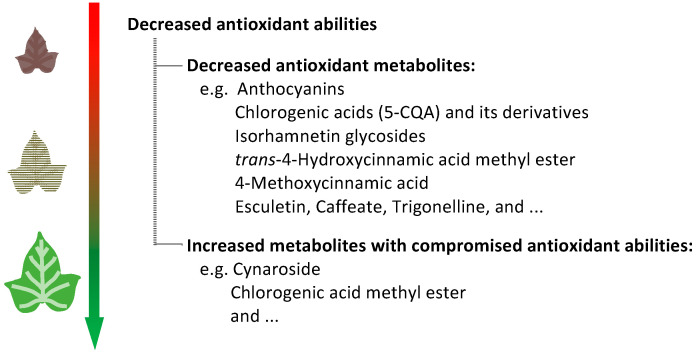
Scheme for the decreased antioxidant abilities in sweetpotato JRF leaves.

**Table 1 plants-11-03014-t001:** The 449 metabolites identified in sweetpotato leaves of “Chuanshan Zi”.

Category (Compound Number)		Class No.	Class	Compound Number	Correlation with Antioxidant Abilities
Primary metabolites (233)		Class 1	Amino acid and derivatives	55	0.477
		Class 2	Nucleotide and derivatives	41	0.955 **
		Class 3	Lipids	52	−0.418
		Class 4	Carbohydrates	9	0.759
		Class 5	Organic acids and derivatives	47	−0.938 *
		Class 6	Vitamins and derivatives	14	0.769
		Class 7	Indole derivatives	6	0.003
		Class 8	Alcohols	9	0.971 **
Secondary metabolites (193)					
Alkaloids (18)		Class 9	Alkaloids	18	0.877
Terpenes (8)		Class 10	Terpenes	8	0.205
Phenolics (167)					
	Simple Phenols (82)	Class 11	Phenylpropanoids	46	0.944 *
		Class 12	Phenolic acids	25	−0.966 **
		Class 13	Phenolamides	9	0.937 *
		Class 14	Quinones	2	−0.963 **
	Flavonoids (85)	Class 15	Flavonoids	12	0.578
		Class 16	Flavanones	10	0.192
		Class 17	Flavones	23	−0.578
		Class 18	Isoflavones	3	−0.852
		Class 19	Flavonols	21	0.846
		Class 20	Anthocyanins	5	0.776
		Class 21	Proanthocyanidins	4	0.956 **
		Class 22	Polyphenols	7	0.251
Others (23)		Class 23	Others	23	0.734

Note: Correlation significance were shown at the 0.01 level (**) and 0.05 level (*).

**Table 2 plants-11-03014-t002:** The top 50 abundant compounds ranked by the average peak areas of all samples.

Rank	Compounds	Class	Average Peak Area (×10^7^)	Rank	Compounds	Class	Average Peak Area (×10^7^)
No.1 ^#^	Cynaroside (Luteolin 7-*O*-glucoside)	Class 17	8.47 ± 2.34	No.26	MAG (18:3) isomer2	Class 3	2.07 ± 0.14
No.2	L(-)-Malic acid	Class 5	6.82 ± 0.54	No.27	6,7-Dihydroxycoumarin 7-*O*-quinic acid	Class 11	2.03 ± 0.22
No.3	L-Leucine	Class 1	5.49 ± 0.15	No.28 *	Isorhamnetin *O*-hexoside	Class 19	1.97 ± 0.65
No.4	MAG (18:3) isomer3	Class 3	5.20 ± 0.35	No.29 *	Isorhamnetin 5-*O*-hexoside	Class 19	1.91 ± 0.62
No.5	alpha-Aminocaproic acid	Class 1	5.09 ± 0.17	No.30	Procyanidin B3	Class 21	1.83 ± 0.33
No.6 ^#^	Kynurenic acid	Class 5	4.38 ± 1.67	No.31	Glutamic acid	Class 1	1.82 ± 0.20
No.7	Citric acid	Class 5	3.71 ± 0.38	No.32 ^#,^*	Trigonelline	Class 9	1.81 ± 0.34
No.8	LysoPC 20:4	Class 3	3.64 ± 0.40	No.33	Peonidin 3-*O*-glucoside	Class 20	1.81 ± 0.22
No.9	Niacinamide	Class 6	3.57 ± 0.32	No.34	2-Isopropylmalate	Class 5	1.80 ± 0.15
No.10	2-Aminoisobutyric acid	Class 1	3.55 ± 0.47	No.35 ^#,^*	Caffeate	Class 11	1.76 ± 0.79
No.11 ^#^	Galactinol	Class 4	3.37 ± 0.75	No.36 ^#^	Cyanidin 3,5-*O*-diglucoside	Class 20	1.67 ± 0.57
No.12	Indole	Class 7	3.30 ± 0.16	No.37 ^#^	LysoPC 20:1	Class 3	1.67 ± 0.38
No.13	L-Phenylalanine	Class 1	3.21 ± 0.19	No.38 ^#^	D-( + )-Sucrose	Class 4	1.63 ± 0.33
No.14	MAG (18:3) isomer5	Class 3	3.10 ± 0.36	No.39 ^#^	MAG (18:4) isomer1	Class 3	1.62 ± 0.38
No.15	Guanosine	Class 2	3.02 ± 0.50	No.40	Peonidin 3-*O*-hexoside	Class 20	1.57 ± 0.30
No.16	Adenosine	Class 2	2.73 ± 0.31	No.41 ^#,^*	4-Methoxycinnamic acid	Class 11	1.56 ± 0.86
No.17 ^#,^*	Chlorogenic acid methyl ester	Class 12	2.73 ± 0.87	No.42	Hesperidin (Hesperetin 7-rutinoside)	Class 16	1.51 ± 0.10
No.18 ^#,^*	*trans*-4-Hydroxycinnamic acid methyl ester	Class 11	2.63 ± 1.46	No.43	Citric acid monohydrate	Class 5	1.48 ± 0.15
No.19 *	D-(-)-Valine	Class 1	2.55 ± 0.23	No.44	LysoPC 15:0	Class 3	1.46 ± 0.18
No.20	14,15-Dehydrocrepenynic acid	Class 3	2.53 ± 0.39	No.45 ^#^	2-Methylsuccinic acid	Class 5	1.46 ± 0.27
No.21 *	Isorhamnetin 3-*O*-glucoside	Class 19	2.42 ± 0.80	No.46 *	Esculetin (6,7-Dihydroxycoumarin)	Class 11	1.42 ± 0.40
No.22	6-MethylCoumarin	Class 11	2.34 ± 0.25	No.47	LysoPC 16:2 (2n isomer)	Class 3	1.42 ± 0.15
No.23 ^#^	Peonidin 3-*O*-sophoroside-5-*O*-glucoside	Class 20	2.26 ± 0.73	No.48 ^#^	LysoPC 18:3	Class 3	1.42 ± 0.54
No.24	2,5-Dihydroxy benzoic acid *O*-hexside	Class 12	2.25 ± 0.41	No.49 ^#^	LysoPC 20:1 (2n isomer)	Class 3	1.42 ± 0.32
No.25	Succinyladenosine	Class 2	2.17 ± 0.22	No.50	Hesperetin 7-*O*-neohesperidoside	Class 16	1.36 ± 0.12

Note: ^#^ indicate compounds that are differentially accumulated in at least one of the comparison groups. * indicate compounds with correlation coefficients (R) that have values |R| > 0.9.

## Data Availability

The metabolic profile is shown in Appendix A.

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
