# Peer review of "Metabolic Profiling and Antioxidant Analysis for the Juvenile Red Fading Leaves of Sweetpotato"

_plants, 2022, doi:10.3390/plants11223014_

Round 1

Reviewer 1 Report

Comments on the MS entitled “Metabolic profiling and antioxidant analysis for the juvenile red fading leaves of sweet potato” authored by Shi et al, bearing MS No. plants-1961113

The MS study conducted by the authors is interesting, following appropriate methodology and analysis. The authors had made adequate effort to prove the hypothesis adopting scientific work flow. The finding would be of interest of readers working in the field of metabolic and MS analysis of plant metabolites. However, the MS could be improved in the lines of the following observations.

It is unclear from the table 2 whether the ranks allotted to the metabolites are for overall growth stages or between any two developmental stage (C1 to C5). According to the Table, Cynaroside has the highest rank in view of its high peak area, similarly for L-malic acid, L-leucine, MAG (18:3) isomer 3, alpha caproic acid, Kynurenic acid. does All these metabolites has ny nutritional value? The merit if the “Chuanshan Zi” to be considered for “functional leafy vegetable” needs to be evaluated in the light of the content of these metabolites, as well.

The leaves from stage 2 to stage 5 of the developmental stage has the highest radical scavenging activity as indicated by the ABTS/DPPH and ferric reducing power. However, the differential metabolites between C2 vs C3, C3 vs C4, C4 vs C5, varies greatly. With C4 vs C5 comprising of highest proportion of downregulated DM. How this can be justified in view of “functional leafy vegetable’s” nutritional value?

In section 2.6, a mere correlation or regression analysis between metabolites and antioxidant capacities would be inconclusive without any information about their antioxidant information background. Whether they are precursor for any functional radical scavenger, or the metabolites themselves are radical scavengers. For eg. The anthocyanins has been claimed in the study to be responsible for antioxidant property of the leaves, however anthocyanins have lower correlation (R=0.779) than Class 2 (Nucleotide and derivatives) and Class 8 (Alcohols) (R > 0.9).

In the discussion, out of 193 metabolites dentified, 86.5% are phenolics, out of which chlorogenic acid as been reported as the main functional component of sweet potato leaves. Does, Chlorogenic acid has antioxidant property? Whereas the study reports flavonoids “Cynaroside” has highest DM peak which is a precursor for lutonin, responsible for high antioxidant activity. Hence, conclusion should be highlighting the significant observations empathically. Also the abundant colourless antioxidants including simple phenols and flavonoids contributing to the high antioxidant activity of the leaves should be the mainstay of the conclusion.

The MS overall should be oriented to highlight the significant metabolites contributing to the high antioxidant property of the sweet potato leaves.

The MS id recommended for “minor revision”

Author Response

Thank you very much for your comments. We have revised the manuscript as suggested. The main changes are as follows:

  1. We added Figure 5 to show the schemefor the decreased antioxidant abilities in sweetpotato JRF leaves.
  2. More details about the Plant materials were added.
  3. Other revisions suggested by reviewers, that can be shown by“Track Changes”

To avoid confusion, we provided the pdf version, with the main changes marked in red.

Point 1: It is unclear from the table 2 whether the ranks allotted to the metabolites are for overall growth stages or between any two developmental stage (C1 to C5).

Response 1: It is for overall growth stages, as stated at the title of Table 2 (Line 199, 205)

Point 2: According to the Table, Cynaroside has the highest rank in view of its high peak area, similarly for L-malic acid, L-leucine, MAG (18:3) isomer 3, alpha caproic acid, Kynurenic acid. does All these metabolites has any nutritional value? The merit if the “Chuanshan Zi” to be considered for “functional leafy vegetable” needs to be evaluated in the light of the content of these metabolites, as well.

Response 2: Metabolomic data mainly refers to endogenous small molecules with a relative molecular weight less than 1000 (Line 95), and are more used for identification of secondary metabolites. Nutrition from macro-molecules, such as proteins and fibers, were not shown by metabolic data. Primary metabolites were mainly the intermediate products of primary metabolism. They do possess nutritional value, but are less attributed as functional molecules. Among the top 20 compounds, 17 were primary metabolites, indicating their higher abundance over secondary metabolites. Then as the rank goes down, the proportion of secondary metabolites increased.

Point 3: The leaves from stage 2 to stage 5 of the developmental stage has the highest radical scavenging activity as indicated by the ABTS/DPPH and ferric reducing power. However, the differential metabolites between C2 vs C3, C3 vs C4, C4 vs C5, varies greatly. With C4 vs C5 comprising of highest proportion of down-regulated DM. How this can be justified in view of “functional leafy vegetable’s” nutritional value? 

Response 3: The antioxidant capacities showed a descending order of C1 > C2 > C3 > C4 > C5 (Line 90-91). The DMs reflect not only antioxidants, but also metabolite alternation during leaf development. Our former transcriptomic analysis also showed highest proportion of down-regulated genes in C4-vs-C5, consistent with the metabolic data. These data indicate many metabolites have higher contents in young leaves than mature leaves (revised in Line 176-177).

Point 4: In section 2.6, a mere correlation or regression analysis between metabolites and antioxidant capacities would be inconclusive without any information about their antioxidant information background. Whether they are precursor for any functional radical scavenger, or the metabolites themselves are radical scavengers. For eg. The anthocyanins has been claimed in the study to be responsible for antioxidant property of the leaves, however anthocyanins have lower correlation (R=0.779) than Class 2 (Nucleotide and derivatives) and Class 8 (Alcohols) (R > 0.9).

Response 4: We checked the background for many of the secondary metabolites, with most of them reported to have some health function including antioxidant ability. We analyzed the metabolic data for a long time, trying to explore some specific compounds instead of sinking into general and obscure conclusions. The correlation analysis were adopted, although it has some drawbacks.

The antioxidant background for the highlighted compounds were included in the three sections of Discussion.

Primary metabolites were mainly the intermediate products of primary metabolism (revised in Line 100). They do not possess such strong antioxidant background, and were not specially discussed. The high correlation for Class 2 and Class 8 (belonging to primary metabolites) may indicate some possible associations among these metabolic pathways.

 Point 5: In the discussion, out of 193 metabolites dentified, 86.5% are phenolics, out of which chlorogenic acid as been reported as the main functional component of sweet potato leaves. Does, Chlorogenic acid has antioxidant property? Whereas the study reports flavonoids “Cynaroside” has highest DM peak which is a precursor for luteolin, responsible for high antioxidant activity. Hence, conclusion should be highlighting the significant observations empathically. Also the abundant colourless antioxidants including simple phenols and flavonoids contributing to the high antioxidant activity of the leaves should be the mainstay of the conclusion.

Response 5: Thank you for your advise. We added figure 5 and rewrote the conclusion.

Chlorogenic acid (5-CQA) and its derivatives (Figure 4B) have good antioxidant property , and were discussed in more details in “3.2 Key compounds...” (Line 310-315). “Cynaroside” was also highlighted as suggested (Line 264-266, Figure 5).

Point 6: The MS overall should be oriented to highlight the significant metabolites contributing to the high antioxidant property of the sweet potato leaves.

Response 6: Thank you for your advise. We revised the Abstract and Conclusion to highlight these significant metabolites. We tried to discuss these compounds in the Discussion, which made the discussion too lengthy to show the points.

However, we also wish to provide some clues for the debate of “functions of anthocyanins in juvenile leaves”, which is an important ecological question.

Reviewer 2 Report

The manuscript “Metabolic profiling and antioxidant analysis for the juvenile red fading leaves of sweetpotato” by Shi, et al., who done the metabolomic profiling and antioxidants analysis at five leaf stages for the sweetpotato cultivar “Chuanshan Zi”. This manuscript is well written and thoroughly described and it can find interest among the researchers in this field.

The authors present the results of probably one experiment, determining the metabolomic profiles and antioxidant capacity of sweetpotato. Considering the environmental factors for such kind of research, the results/data collected from more seasons in more orchards are more reliable than those from one season within one orchard. I think the data presented here is not justifiable to support the objectives of this study.

Other minor suggestions are below as well

In the abstract section the authors should also include some key findings rather than the concluding remarks. Also, in the last, authors are suggested to highlight the novelty of the manuscript.

Why sweetpotato cultivar “Chuanshan Zi” was selected for the study? Any specific reason?

Improve discussion by providing some scheme of the responsible mechanisms for the variations.

Considering the environmental factors such as temperature, light etc the authors should provide some information regarding these factors in the introduction section. Please consult the following bibliography

Zhang L, Gao Y, Deng B, Ru W, Tong C and Bao J (2022) Physicochemical, Nutritional, and Antioxidant Properties in Seven Sweet Potato Flours. Front. Nutr. 9:923257. doi: 10.3389/fnut.2022.923257

Yan C, Zhang N, Wang Q, Fu Y, Wang F, Su Y, Xue B, Zhou L and Liao H (2021) The Effect of Low Temperature Stress on the Leaves and MicroRNA Expression of Potato Seedlings. Front. Ecol. Evol. 9:727081. doi: 10.3389/fevo.2021.727081

Silvestri C, Caceres ME, Ceccarelli M, Pica AL, Rugini E and Cristofori V (2019) Influence of Continuous Spectrum Light on Morphological Traits and Leaf Anatomy of Hazelnut Plantlets. Front. Plant Sci. 10:1318. doi: 10.3389/fpls.2019.01318

Yu J, Su D, Yang D, Dong T, Tang Z, Li H, Han Y, Li Z and Zhang B (2020) Chilling and Heat Stress-Induced Physiological Changes and MicroRNA-Related Mechanism in Sweetpotato (Ipomoea batatas L.). Front. Plant Sci. 11:687. doi: 10.3389/fpls.2020.00687

I think both morphological and physiological parameters should also have been measured using appropriate techniques.

There are many confusions in the M&M. It looks like the authors have intermingled a lot of things together and they are lacking the symmetry of the MS. Please provide a detailed description of how the experiments were assumed. How many combinations were there in each experience? How many plants were in one repetition. How the plants were sampled for analysis and measurements. Please complete this information in the individual sections of the Materials and Methods chapter

The results and discussions are too lengthy which is a good thing but on the same note it makes the justifications for the results a little more exaggerated. The authors should stick to the precise and strong justifications to address their objectives achieved in this study. Moreover, most of the literature is outdated, that should be checked as well to add some latest findings to support your arguments. Furthermore, there are many language and spelling errors in this section as well.

To deepen in the conclusion to highlight further aspects. Recommendations for future studies for other regions and for other types of crops should also be included. Even under other production systems.

The references are not properly formatted. Please follow the author’s guidelines provided by the journal.

Author Response

Thank you very much for your comments. We have revised the manuscript as suggested. The main changes are as follows:

  1. We added Figure 5 to show the schemefor the decreased antioxidant abilities in sweetpotato JRF leaves.
  2. More details about the Plant materials were added.
  3. Other revisions suggested by reviewers, that can be shown by“Track Changes”

To avoid confusion, we provided the pdf version, with the main changes marked in red.

Point 1: The authors present the results of probably one experiment, determining the metabolomic profiles and antioxidant capacity of sweetpotato. Considering the environmental factors for such kind of research, the results/data collected from more seasons in more orchards are more reliable than those from one season within one orchard. I think the data presented here is not justifiable to support the objectives of this study.

Response 1: We add details about leaf samples in M&M section. The cultivar “Chuanshan Zi” is widely distributed and planted. We have planted the cultivar from 2017 to 2022, and it would show repeatable phenotype under adequate nutrition and light with seasonal temperature from 15℃ to 35℃.

The measurements of “anthocyanins, TPC, TFC, and antioxidant activities ” were done several times in different years, and showed similar results. In order to make the correlation results between antioxidant and metabolites more reliable, the leaf samples used in this study are were collected between May to July, 2020. We analyzed the metabolic data for a long time, trying to explore some specific compounds instead of some general and obscure conclusion.

Point 2: In the abstract section the authors should also include some key findings rather than the concluding remarks. Also, in the last, authors are suggested to highlight the novelty of the manuscript.

Response 2: Thank you for your advise. We revised the abstract as suggested.

Point 3: Why sweetpotato cultivar “Chuanshan Zi” was selected for the study? Any specific reason?

Response 3: Reason added (Line 60-62).

Point 4: Improve discussion by providing some scheme of the responsible mechanisms for the variations.

Response 4: Thank you for your advise. Figure 5 were added.

Point 5: Considering the environmental factors such as temperature, light etc the authors should provide some information regarding these factors in the introduction section. Please consult the following bibliography

Response 5: Yes, you are right. Environmental factors can influence plant phenotype greatly. During leaf sampling, we avoided the variation caused by external environmental factors, and only used vines with repeatable phenotype.

As it is hard to explain external factors in one or two sentences, the suggested information were added in the M&M part (Line 370-382) to avoid interference to the content continuity of the Introduction.

Point 6: I think both morphological and physiological parameters should also have been measured using appropriate techniques.

Response 6: This is a follow-up study of our former research (Deng et al. 2020), which described some morphological and physiological parameters, and also the transcriptomic analyses for leaf development.

Point 7: There are many confusions in the M&M. It looks like the authors have intermingled a lot of things together and they are lacking the symmetry of the MS. Please provide a detailed description of how the experiments were assumed. How many combinations were there in each experience? How many plants were in one repetition. How the plants were sampled for analysis and measurements. Please complete this information in the individual sections of the Materials and Methods chapter

Response 7: Thank you for your advice. Mere citation of our former research caused confusions. We added the details of plant sampling (Line 381-386), which were formerly simplified to reduce the length of the MS.

Point 8: The results and discussions are too lengthy which is a good thing but on the same note it makes the justifications for the results a little more exaggerated. The authors should stick to the precise and strong justifications to address their objectives achieved in this study. Moreover, most of the literature is outdated, that should be checked as well to add some latest findings to support your arguments.

Response 8: Thank you for your advice. Although we should be oriented to highlight the significant metabolites contributing to alteration of antioxidant property in sweetpotato leaves, we also wish to provide some clues for the debate of “functions of anthocyanins in juvenile leaves”, which is an important ecological question. The above considerations make the discussion too lengthy.

Point 9: Furthermore, there are many language and spelling errors in this section as well.

Response 9: Thank you for your advice. We thoroughly checked the manuscript to correct the problem.

Point 10: To deepen in the conclusion to highlight further aspects. Recommendations for future studies for other regions and for other types of crops should also be included. Even under other production systems.

Response 10: Thank you for your advise. We rewrote the conclusion..

Point 11: The references are not properly formatted. Please follow the author’s guidelines provided by the journal.

Response 11: Thank you for your advise. We reformatted the references with the “Endnote module for MDPI”.

Reviewer 3 Report

The subject of the manuscript is relevant and original. It is suitable for the Plants Journal. Due to the conducted experiment, the article is very original, contains significant scientific value, and the results should be promulgated in the pharmaceutical and food industries.

Abstract:

The abstract seems very cursory. I suggest adding information about the year and location of the experiment.

Introduction:

The introduction presents the current state of knowledge on this subject but there is no mention of the purpose of the paper. Please correct that.

Materials and Methods

The materials and methods lack the description of the dates when the leaf samples were collected for the tests and the year of the study. The fragment mentioning only the author of the paper in which the research methodology is explained is very cursory. Please correct that. There are no reservations to the statistical methods, which are suitable.

Results and Discussion

There are no major reservations to the results. The results are complete and appropriately described.

Conclusions

The conclusions are too general, they should contain a more detailed summary of the research results. Perhaps, you could also mention that the research shall be continued.

References

The references are well selected and their style is consistent with the journal requirements.

Subject to the additions, the paper can be published in the Plants Journal.

Author Response

Thank you very much for your comments. We have revised the manuscript as suggested. The main changes are as follows:

  1. We added Figure 5 to show the schemefor the decreased antioxidant abilities in sweetpotato JRF leaves.
  2. More details about the Plant materials were added.
  3. Other revisions suggested by reviewers, that can be shown by“Track Changes”

To avoid confusion, we provided the pdf version, with the main changes marked in red.

Point 1: Abstract: The abstract seems very cursory. I suggest adding information about the year and location of the experiment.

Response 1: The information about the year and location of the experiment is lengthy. We add the suggested details in M&M section (Line 370-378). The cultivar “Chuanshan Zi” is widely planted. We have planted the cultivar from 2017 to 2022, and it would show repeatable phenotype under adequate nutrition and light with the seasonal temperature from 15℃ to 35℃. The assays of “anthocyanins, TPC, TFC, and antioxidant activities measurements” were done several times across different years, and showed similar results. In order to make the correlation results between metabolites and antioxidant ability more reliable, the leaf samples used in this study are were collected between May to July, 2020.

Point 2: Introduction: The introduction presents the current state of knowledge on this subject but there is no mention of the purpose of the paper. Please correct that.

Response 2: Revised as suggested (Line 64-65).

Point 3: Materials and Methods: The materials and methods lack the description of the dates when the leaf samples were collected for the tests and the year of the study. The fragment mentioning only the author of the paper in which the research methodology is explained is very cursory. Please correct that. There are no reservations to the statistical methods, which are suitable.

Response 3: Revised as suggested. More details were added in M&M section (Line 370-378)

Point 4: Conclusions: The conclusions are too general, they should contain a more detailed summary of the research results. Perhaps, you could also mention that the research shall be continued.

Response 4: Thank you for your advise. We rewrote the conclusion.

Round 2

Reviewer 2 Report

Dear Authors, 

I really appreciate your efforts in improving the manuscript. 

Best regards